# Transformation of Multi-Antibiotic Resistant *Stenotrophomonas maltophilia* with GFP Gene to Enable Tracking its Survival on Pine Trees

**Yu-Long Li, Cui-Yun Zheng, Kan-Cheng Liu, Yang Wu, Ben Fan and Zheng-Min Han ***

Co-Innovation Center for Sustainable Forestry in Southern China, College of Forestry,
Nanjing Forestry University, Nanjing 210037, China; lyj.lylo@163.com (Y.-L.L.); 15077876359@163.com (C.-Y.Z.);
liukancheng@163.com (K.-C.L.); wuyang210017@163.com (Y.W.); fanben2000@gmail.com (B.F.)
* Correspondence: zmhan@njfu.edu.cn; Tel.: +86-025-8542-7397

**Abstract:** *Pinus massoniana* Lamb., commonly known as Masson Pine, is one of the most important tree species for planted forests in China. This species is, however, threatened by pine wilt disease caused by *Bursaphelenchus xylophilus*. *Stenotrophomonas maltophilia* (Palleroni & Bradbury 1993) Smal-007, a bacterium isolated from the body surface of native *B. xylophilus*, was evidenced to possess the ability to prevent and control this disease. In this study, we focus on exploring effective transformation and green fluorescent protein (GFP)-labeling of Smal-007, in order to facilitate its later investigation. The results indicated that the recombination of antibiotic Tp (trimethoprim), and the uncoupling reagent, CCCP (carbonyl cyanide m-chlorophenyl hydrazine), was effective for the transformation of the multidrug-resistant bacterium. An optimal transformation procedure, including electroporation, was established. To the best of our knowledge, this is the first report where such a method was used for *S. maltophilia* transformation. Furthermore, Smal-007 was labeled by GFP, allowing the monitoring of its survival ability in pine trees. The labeling was robust and recognizable in isolates recovered from pine needles and bark. In summary, our study indicated that combining uncoupling reagents could be a useful approach to finding operative antibiotic markers for the transformation of multidrug-resistant bacteria. In addition, our successful labeling of Smal-007 with GFP could improve the understanding of its ecological impact, when used as a biocontrol agent.

**Keywords:** *Stenotrophomonas maltophilia*; antibiotic resistance; transformation; uncoupler; GFP; pine wilt disease

## 1. Introduction

*Stenotrophomonas maltophilia* is a Gram-negative bacterium, which is widely distributed in various environments of nature. *S. maltophilia* can biodegrade alkane and polycyclic aromatic hydrocarbons (PAHs), and absorb heavy metals, and therefore, plays a critical role in the ecological cycle. *S. maltophilia* is also a typical plant growth-promoting rhizobacterium. It can secrete various secondary metabolites that enhance plant growth, prevent and control fungal diseases. These characteristics endow *S. maltophilia* with great application potential in environmental remediation, agricultural production and forest protection [1–3].

Pine wilt disease is the most dangerous pine disease in East Asia [4,5]. At present, there is no effective prevention and curative method for this disease. Pine wilt disease is generally regarded to be caused by pine wood nematode *Bursaphelenchus xylophilus*, which originates from North America [6]. While some scientists believe that *B. xylophilus* itself can directly result in pine wilt disease, others attribute this disease to bacteria carried by *B. xylophilus* [7–11]. *S. maltophilia* Smal-007 is an epiphytic

bacterium isolated from nematodes derived from the USA, and has shown an ability to lower the incidence of pine wilt disease [9,12]. The mechanism of the biocontrol activity of Smal-007 against pine wilt disease is not completely clear. Since Smal-007 does not antagonize *B. xylophilus*, a hypothesis was raised that non-toxic Smal-007 may replace those *B. xylophilus*-associated bacteria with high toxicity to pines [9,12,13]. Probably due to long adaption with nematodes, Smal-007 is more competent to dominate surfaces of *B. xylophilus*, than the pathogenic bacteria. This replacement may thus significantly reduce the pathogenicity of *B. xylophilus* to pines [12].

In order to study the colonization ability of Smal-007 as a biocontrol agent on pine trees, it is necessary to find an effective marker. Rifampicin resistance screening, enzyme-linked immunosorbent assay (ELISA), and immunolabeling techniques have been used, but proved to be incompetent in tracing the strain effectively [14,15]. Green fluorescent protein (GFP) labeling was successful in monitoring microbes in different environments [16,17]. Therefore, GFP labeling is an ideal choice, worth trying to track Smal-007.

Labeling bacteria with a marker generally needs an antibiotic-resistant gene during transformant screening. Some bacteria, however, in nature have multidrug resistance, such as *Staphylococcus aureus*, *Pseudomonas fluorescens*, *Burkholderia cepacia* and *S. maltophilia* [18–20]. Multidrug resistance is often caused by catabolic enzymes or/and efflux pumps in the bacterial membrane. For transformation, the multidrug-resistant strains were usually mutated to acquire specific antibiotic sensitivity, as is, however, time-consuming and in many cases inefficient [21]. *S. maltophilia* is a typical bacterium with multidrug resistance, a bottleneck for its transformation. In this study, we screened multiple antibiotics and found that one of them, coupled with uncoupling reagent, is an effective tool for transformation. An optimal transformation procedure was established, which theoretically could be generalized and applied to other multidrug-resistant bacteria. With this method, we successfully labeled Smal-007 with GFP. The GFP-marked Smal-007 could be helpful in studying the mechanisms of how it controls pine wilt disease.

## 2. Materials and Methods

### 2.1. Bacterial Strains, Plasmids

*S. maltophilia* Smal-007 was isolated from nematodes derived from the USA. Luria–Bertani (LB) broth contained 1% (W/V) tryptone (OXOID), 0.5% (W/V) yeast extract (OXOID), and 0.5% (W/V) NaCl. The media was solidified through the addition of agar to 1.5%. The SOC (Super Optimal broth with Catabolite repression) medium contained 2% (W/V) tryptone, 0.5% (W/V) yeast extract, 0.05% (W/V) NaCl, 2.5 mM KCl 10 mM $MgCl_2$, 20 mM glucose. The antibiotics used for *Escherichia coli* were spectinomycin (100 µg/mL), chloramphenicol (34 µg/mL), kanamycin (50 µg/mL), and ampicillin (100 µg/mL). The strains and plasmids used are listed in Table 1.

**Table 1.** Strains and plasmids.

| Category | Feature | Source or Reference |
|---|---|---|
| Plasmids | | |
| pMD-19-Tp | pMD-19, $Tp^R$ | This work |
| pBBR1-Tp | Extensive host vectors, TpR | [22] |
| pBBR1-TpGFP | Extensive host vectors with *gfp*, $Tp^R$ | Laboratory stock |
| pMD-19-xanGFP | *gfp* and $tp^R$ were inserted in the intergenic region between *xanA and xanB* gene | This work |
| Strains | | |
| *E.coli* Top10 | | Laboratory stock |
| *Stenotrophomonas maltophilia* Smal-007 | Multidrug-resistant | Laboratory stock |
| Smal-007GFP | With plasmid pBBR1-TpGFP | This work |
| Smal-007GFPI | Transformant of pMD-19-xanGFP | This work |

$Tp^R$, trimethoprim resistance. *gfp*: green fluorescence protein.

## 2.2. Antimicrobial Minimum Inhibitory Concentration (MIC) Test

After incubation on LB agar at 30 °C for 24 h, a colony of the Smal-007 strain was transferred to LB medium and cultivated until $OD_{600}$ (optical density) reached about 1.0. For each sample, a culture of 500 μl was spread on LB plates with a different concentration of an antibiotic. After incubation at 30 °C for 24 h, the MIC (antimicrobial minimum inhibitory concentration) was determined as corresponding to the antibiotic concentration of plates without clone growth. Based on the uncoupler minimum bactericidal concentration (MBC) test of CCCP (carbonyl cyanide m-chlorophenyl hydrazine) from 2.5 μg/mL to 30 μg/mL, the MIC of complex use of antibiotics and CCCP were tested in the same method [23]. In total, the MIC of 11 antibiotics were tested in the same method. The gradient of the MIC test of antibiotics was 100 μg (100 μg-1000 μg).

## 2.3. Preparation of Competent Cells for Electroporation

Competent cells of *S. maltophilia* Smal-007 were prepared as previously described. In brief, Smal-007 was grown in 10 mL of LB medium at 30 °C and 200 rpm until $OD_{600}$ reached 1.0. Then all the culture was cooled on ice for 30 min. The cells were collected by centrifugation at 4 °C, 8000 rpm for 1 min and were subsequently re-suspended in chilled EWB (10% redistilled glycerol in ultrapure water). Next, the cells were centrifuged at 4 °C, 8000 rpm for 1 min again, before the supernatant was discarded. The pellet was washed twice by EWB and re-suspended by chilled EWB for use [24,25].

## 2.4. Electroporation

An aliquot of competent cells of 50 μl was chilled in polypropylene tubes, before that, 1-μg DNA was added and mixed thoroughly. Then the mixture was transferred to a chilled 0.1 cm electroporation cuvette. The electroporation was performed at 25 kV, 25 μF and 200 Ω. The shocked cells were immediately moved from the cuvette to a microcentrifuge tube. The SOC medium was added to bring the final volume to 1 mL after shaking at 200 rpm, and 30 °C for 3 h. The cells were spread on a medium, supplemented with antibiotics and grown at 30 °C for 48 h [26].

## 2.5. Labelling of S. Maltophilia Smal-007 with GFP

The *E. coli* strain Top10 and pMD-19-T were used to construct the recombinant vector. The intergenic region between *xanA* and *xanB* of the Smal-007 genome was selected as the insert site of GFP. The upstream and downstream homologous fragments were amplified using Smal-007 genome as a template and the primer pairs of xan_upfr: (CCCAAGCTTCGACCGATGAG CGGGCCC) plus xan_uprev (CCCAAGCTTCGCATGAGCAGCATCCAGC), and Xan_dwfr (GGGGTACCGGCCTCGGTTACTGCTGG) plus Xan_dwrev (GGGGTACCGGACTGGGACTACAATCG), respectively. The amplified fragments were inserted into the pMD-19 after digestion by Hind III and Kpn I. The resulting pMD-19-Xan Tp (trimethoprim) and GFP was amplified using pBBR1-TpGFP as the template and a pair of primers: GFP_fr (CTCTAGAGTTTGGGCAACAAACTAATGT) and tp_rev (GCTCTAGACATTCTTGCCCGCCTGAT). The PCR product of GFP-Tp was inserted to pMD-19-Xan after digestion by Xba I. The yielded plasmid was transformed into Smal-007 as described above.

The stability of the GFP tag was tested by continuous streaking on LB agar. In brief, a single colony of the GFP-tagged strain was streaked on LB agar. After incubation at 30 °C for 48 h, a single fluorescence colony was picked and streaked on a second LB agar. The control was Smal-007 carrying pBBR1-TpGFP. The fluorescence of the colonies was recorded by the imager Fluorchem Q (Protein Simple, USA).

## 2.6. Recovery Experiment of GFP-Tagged Smal-007 from Pine Trees

A single colony of Smal-007 was incubated in LB medium by shaking (200 rpm) at 30 °C for 48 h. The culture was diluted to $1 \times 10^8$ CFU/mL and sprayed on pine needles and bark of 20 trees grown in a closed experimental station in JuRong, JiangSu province, until the needles and the bark were fully moist. Medium with the absence of Smal-007 was used as the negative control.

Pine needles and bark were collected randomly from the sprayed pines. The needles of 1 gram and the bark of 0.1 gram of each sample were soaked in 1 mL sterile saline for one hour. The soak solution of 500 μl was 10 times continuously diluted and plated on LB plates with Tp. The plates were incubated at 30 °C for 48 h and the GFP-tagged colonies were counted.

## 3. Results

### 3.1. Uncoupler Minimum Bactericidal Concentration (MBC) Test

A high dose of the uncoupler CCCP may interfere with cell metabolism of *S. maltophilia* [27], therefore its MBC was firstly determined before use. The result (Figure 1) shows that the Smal-007 survival rate was not affected when the CCCP concentration was lower than 10 μg/mL. The strain survival rate dropped sharply when the CCCP concentration was higher than 10 μg/mL and no strain grew at the corresponding plate of 20 μg/mL. Therefore, 10 μg/mL was the maximum concentration of decreasing antibiotics export of Smal-007, on the premise that non-interference happens to normal metabolism. We thus decided to use a CCCP concentration of 8 μg/mL for subsequent experiments.

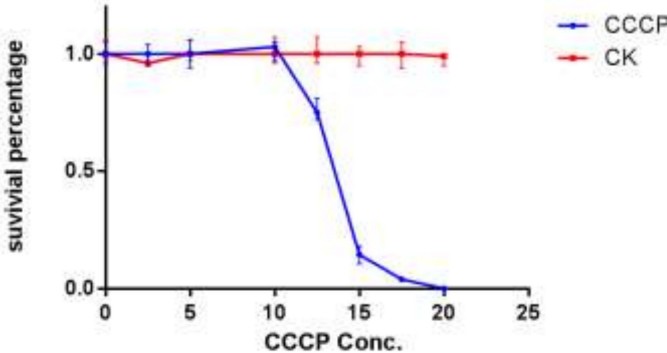

**Figure 1.** Minimum bactericidal concentration test of the uncoupler CCCP (carbonyl cyanide m-chlorophenyl hydrazine) to Smal-007. The survival rate of *Stenotrophomonas maltophilia* Smal-007 after 48 h of exposure to CCCP at different concentrations were recorded. CK represents survival rate of control group without CCCP. X-axis represents concentration of CCCP (μg/mL) and Y-axis represents the survival rate of Smal-007.

### 3.2. Antimicrobial Minimum Inhibitory Concentration (MIC) Test

The sensitivity of *S. maltophilia* Smal-007 for a total of 11 frequently used antibiotics was tested. The result showed that Smal-007 was not sensitive to any of the antibiotics under 400 μg/mL (Figure 2A). It was relatively more sensitive to the rifampicin, chloramphenicol and erythromycin, compared to other antibiotics. The MIC of trimethoprim, streptomycin, spectinomycin, ampicillin, lincomycin and kanamycin was out of the range of the tested maximum concentration (1000 μg/mL). This result proved that Smal-007 has multidrug resistance, similar to the previous reports of other *S. maltophilia* strains [20,28].

The MIC of 11 antibiotics (100 to 400 μg/mL) was tested when used in combination with CCCP at 8 μg/mL. It was noted that CCCP lowered the resistance of Smal-007 to trimethoprim by ~60% (Figure 2B). This indicated that the resistance of Smal-007 to trimethoprim was mainly caused by efflux pumps. Therefore, we decided to choose a trimethoprim concentration of 800 μg/mL as a selective condition for the subsequent Smal-007 transformation.

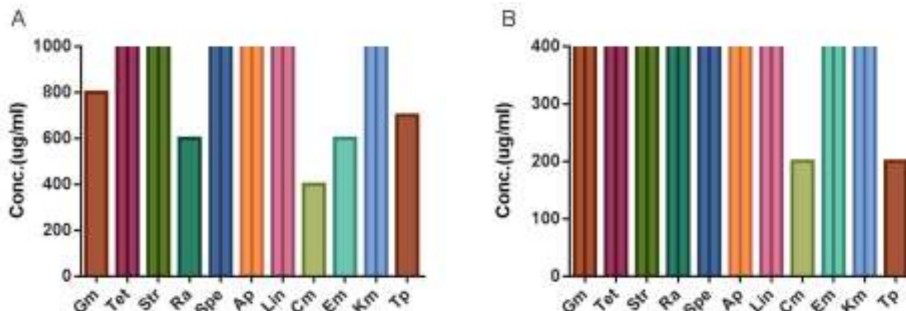

**Figure 2.** The minimum inhibitory concentration (MIC) test of Smal-007. (**A**) MIC of 11 antibiotics. (**B**) MIC of 11 antibiotics with the uncoupler CCCP. Smal-007, *Stenotrophomonas maltophilia* Smal-007. CCCP, carbonyl cyanide m-chlorophenyl hydrazine. 11 antibiotics, Gm: gentamicin, Str: streptomycin, Spe: spectinomycin, Ap: ampicillin, Lin: lincomycin, Cm: chloramphenicol, Em: erythrocin, Km: kanamycin, Tp: trimethoprim. Y-axis represents concentration of the antibiotics used. Line throughout the column represents that the MIC was out of range of the maximum test concentration.

### 3.3. Transformation of S. Maltophilia Smal-007

Three common transformation methods: Freeze-thawing in liquid nitrogen, the CaCl$_2$ method and electroporation, were tested to individually transform two plasmids with the trimethoprim resistance gene into Smal-007. Only electroporation successfully yielded Smal-007 transformants (Figure 3).

No satellite colonies were observed around the transformants after 48 h. Multiple colonies were picked and observed under 475 nm light of a fluorescence microscope (Axio Scope A1, Carl Zeiss, Gottingen, Germany). All picked colonies contained cells with bright green fluorescence, indicating successful transformation.

A high voltage and higher plasmid concentration can improve the transformation efficiency. The optimized transformation mixture included 5 μl DNA solutions (the concentration was higher than 500 ng/ul) and 50 μl Smal-007 competent cells. The electric shock condition used was 200 Ω, 25 μF at 2.5 kv. After shock, 1 mL of SOC medium was used to recover the cells, by shaking at 200 rpm and a 30 °C incubation for 3 h. The transformants were selected on LB agar containing trimethoprim at 800 μg/mL. With this method, we successfully transformed pMD-19-xanGFP and pBBR1-TpGFP into Smal-007, separately. The trimethoprim resistance gene and GFP gene were integrated into the intergenic region of *xanA* and *xanB*, as verified by PCR and sequencing.

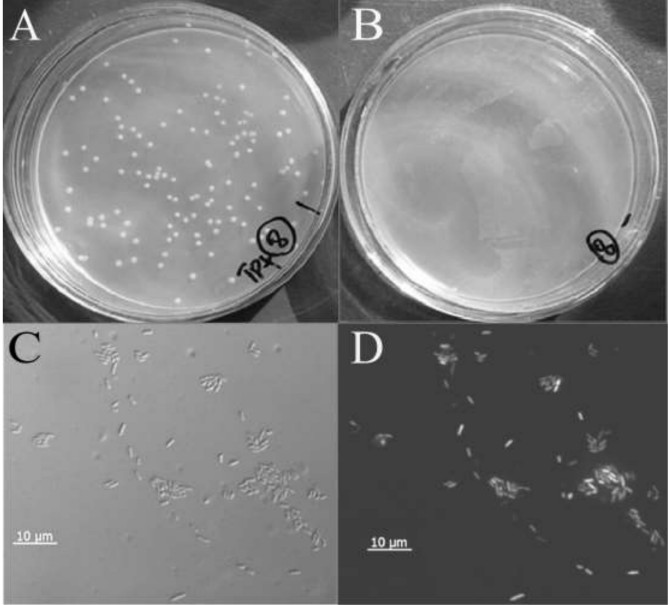

**Figure 3.** Colonies and microscopic observation of Smal-007 transformed by pBBR1-TpGFP. (**A**) The transformants growth on LB plate. (**B**) Negative control. (**C–D**) Fluorescence microscopy observation of the transformants. C: light field; D: UV (475cnm) field. Smal-007, *Stenotrophomonas maltophilia* Smal-007.

### 3.4. Stability Tests of GFP Tag

The GFP stability of two types of transformants, Smal-007GFP (Smal-007 labeled by *gfp*-containing pBBR1-TpR) and Smal-007GFPI (Smal-007 labeled by integrated *gfp*), were compared. The percentage of fluorescent cells of Smal-007GFP decreased dramatically from the first plate to the third plate, while all Smal-007GFPI cells kept emitting bright green fluorescence after continuous streaking on three plates (Figure 4). This suggests that the GFP tag was stable in Smal-007GFPI, which therefore may serve as a suitable strain for field investigation of Smal-007.

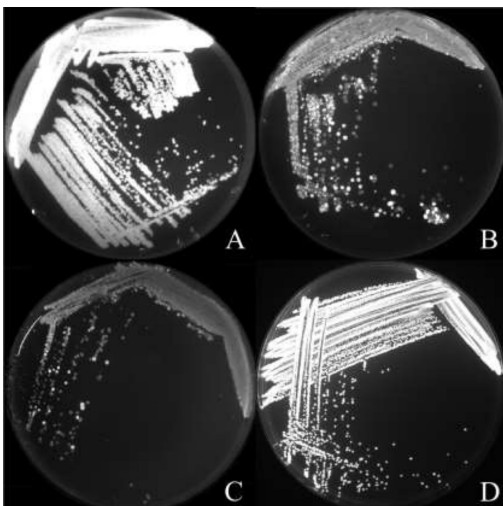

**Figure 4.** Comparison of green fluorescence protein (GFP) fluorescence of *S. maltophilia*. Smal-007 tagged by plasmid (**A, B, C**) and integration on the genome (**D**). A: Smal-007GFP, *S. maltophilia* Smal-007 tagged by gfp-containing plasmid at the first streaking. B: Smal-007GFP at the second streaking. C: Smal-007GFP at the third streaking. D: Smal-007GFPI, *S. maltophilia* Smal-007 with integrated *gfp* at the third streaking.

### 3.5. Colonization of the GFP Marked Strain on The Pine Surface

GFP-labeled Smal-007 was sprayed on pine trees and recovered in one year. The GFP-marked Smal-007 could be easily discriminated from other bacteria recovered, by the clear fluorescence upon excitation (Figure 5A,C). The number of surviving Smal-007 was counted. It was shown that at the initial stage of colonization (the 15th day), Smal-007 recovered from needles and bark were approximately 170 CFU/g and 76,800 CFU/g, respectively. After that, the number of Smal-007 recovered decreased gradually. On the needles, no Smal-007 was recovered on the 75th day (Figure 5B). On the bark, the number of Smal-007 decreased to ~150 CFU/g and kept around this number until the 420th day, which was the end of the test (Figure 5D). These results indicated that Smal-007 could colonize pine bark stably.

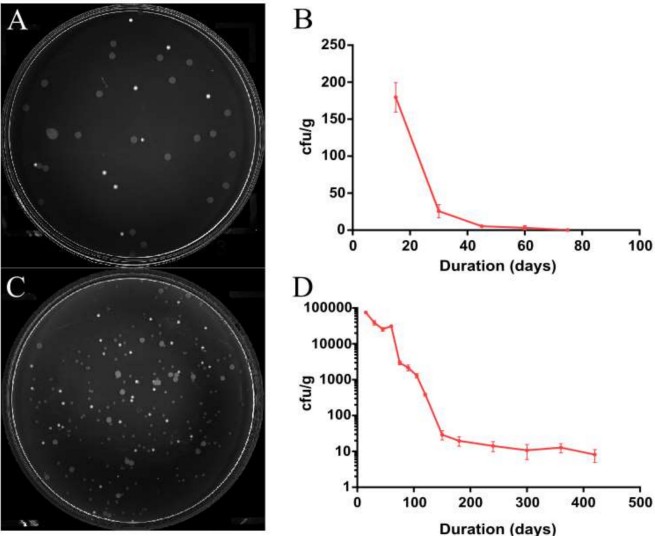

**Figure 5.** *Stenotrophomonas maltophilia* Smal-007 strain isolated from pine trees. (**A**) GFP tagged strain recovered from needles. (**B**) Colonization level of needles. (**C**) GFP tagged strain recovered from bark. (**D**) Colonization level of bark.

## 4. Discussion

In this work, an antibiotic, trimethoprim, was screened out for the transformation of multidrug-resistant *S. maltophilia* Smal-007, together with the uncoupling agent CCCP to inhibit the efflux pump. With trimethoprim resistance as the selective marker, the GFP gene was successfully integrated into the genome of Smal-007. With the GFP label, we monitored Smal-007 survival in pine needles and pine bark.

It is convenient to use Tp and GFP to isolate and identify a target strain from the environment strains. As an effective reporter, GFP greatly facilitated the monitoring of the survival rate of the bacteria Smal-007 in the environment.

The difficulty of transformation is an obstacle to further study multiple antibiotic-tolerant bacteria at the molecular level. Xing Ye screened a chloramphenicol sensitive (MIC 34 μg/mL) strain of *S. maltophilia* and established a genetic transformation method for this strain using chloramphenicol as a selective marker [24]. However, most strains of *S. maltophilia* are Cm-resistant, so Ye's method is not suitable in most cases. Various subspecies of *S. maltophilia* generally have multiple antibiotic resistance by synthesizing antibiotic-detoxifying enzymes, or by reducing the concentration of intracellular antibiotics through efflux pumps on the cytomembrane [29–32]. Antibiotics coupled with uncoupling agents can be an effective method for rapid screening of accessible antibiotics for the transformation of multidrug-resistant bacteria. The successful use of trimethoprim plus CCCP for Smal-007 in this work proved the feasibility of this strategy. Theoretically, this method could be applied to other multidrug-resistant bacteria as well.

　　　The GFP-labeled Smal-007 has a shorter survival term on pine needles. This is probably related to the harsh environmental conditions on needles, such as fewer nutrients, more dynamic temperature, humidity fluctuation and stronger ultraviolet radiation [33,34]. A higher survival rate and long survival term on bark indicted that Smal-007 could effectively colonize pine bark in natural environments, which may be linked to its biocontrol activity against pine wilt disease. In the epidemiological cycle of pine wilt disease, *B. xylophilus* was transmitted by vector beetles such as *Monochamus alternatus* (China, Japan and Korea), *M. carolinensis* (USA) and *M. galloprovincialis* (Portugal) [35–38]. When the beetles move, feed or deposit eggs in pine bark, Smal-007 may have contact with them, and then associate with the vector, *B. xylophilus* [5]. The colonization ability of Smal-007 determines the sustainability of biocontrol, so robust colonization of Smal-007 in pine bark could be a prerequisite of the biocontrol effect. We would use this fluorescent tracer to explore its ecological impact further in future research.

## 5. Conclusions

　　　We successfully labeled the multidrug resistant S. maltophilia Smal-007 with GFP using the combination of the uncoupler CCCP and the antibiotic trimethoprim as a screening marker in the transformation. With the gfp-labeled Smal-007, it was proved that S. maltophilia can colonize pine bark for at least one year.

**Author Contributions:** Z.M.H. and Y.L.L. conceived and designed the project; Y.L.L. and C.Y.Z. undertook the molecular biology experiment; Y.W. and C.Y.Z. recovered the samples; Y.L.L participated in the data analysis and drafted the manuscript; Z.M.H. and B.F. revised and wrote the final version of the manuscript. All authors have read and approved the manuscript for publication.

**Acknowledgments:** This work was supported by National Natural Science Foundation of China (NSFC), Grant/Award Number: 30872026, the Priority Academic Program Development of Jiangsu Higher Education Institutions (PAPD), the Key Scientific Project for Jiangsu Provincial Universities (17KJA220001) and the Agricultural Technology R&D Program of Ju Rong.

**Conflicts of Interest:** The authors declare no conflict of interest.

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
