# Peer review of "Transformation of Multi-Antibiotic Resistant Stenotrophomonas maltophilia with GFP Gene to Enable Tracking its Survival on Pine Trees"

_forests, doi:10.3390/f10030231_

Round 1
Reviewer 1 Report
I enjoyed reading your manuscript entitled – “Transformation of multi-antibiotics resistant Stenotrophomonas maltophilia with GFP gene to enable tracking on pine trees” submitted to Forests.
The study is presented in an appropriate scientific manner and is an extremely valuable piece of work. While this study is of great interest to Microbiologists it also has a broader scope in terms of its application to the pine wilt disease problem which has/is causing huge devastation in China, Japan, South Korea and Portugal. It will be of interest to scientists and foresters directly involved in the study and prevention/control of pine wilt disease.
I have the following comments to make with regards to the manuscript:
The relationship between S. maltophilia Smal-007 and pine wilt disease is stated in lines 35-37. Furthermore the application of the results to pine wilt disease is mentioned very briefly on line 213. Given the significance of these results as a potential biological control for the pine wood nematode it would be very helpful if the authors could give additional detail on the results cited on line 37 and the implications of the results of this study on line 213 in the discussions section.
CCCP is mentioned in the abstract on line 16 but not formally defined until line 67.
Table 1 is referred to in line 61 however I am unable to see any Table 1 in the manuscript.
Many acronyms are defined such as LB and EWB, while others are not. While acronyms like MIC (line 66) and SOC (line 81) are well known amongst microbiologists, given the wider scope of this paper it might be helpful to give the full definitions of these when first mentioned.
S. maltophilia on lines 71, 191, 201, 202 and 125 (there may be other instances too) should be in italics.
B. xylophilus on page 213 should be in italics.
On line 199 Xing Ye is referred to without a reference. It also doesn’t appear to be given in the References section.
There are a number of typos in the manuscript: ‘processe’ on line 48, ‘roducing’ on line 235, on lines 205 and 206 the word ‘of’ is missing before ‘usable antibiotics’ and ‘trimethoprim’ respectively. There are further grammatical errors but these I’m sure will be addressed in the final editing process.
Author Response
Thanks a lot to your valuable commentsThe point-to-point responds to the comments are listed as follows:
Point 1: The relationship between S. maltophilia Smal-007 and pine wilt disease is stated in lines 35-37. Furthermore the application of the results to pine wilt disease is mentioned very briefly on line 213. Given the significance of these results as a potential biological control for the pine wood nematode it would be very helpful if the authors could give additional detail on the results cited on line 37 and the implications of the results of this study on line 213 in the discussions section.
Response 1: We have accordingly added more details in these two places as indicated in the revision.
Point 2: CCCP is mentioned in the abstract on line 16 but not formally defined until line 67.
Response 2: We have defined CCCP in the abstract.
Point 3: Table 1 is referred to in line 61 however I am unable to see any Table 1 in the manuscript.
Response 3: We are sorry for the missing data and now supply Table 1 on line 81.
Point 4: Many acronyms are defined such as LB and EWB, while others are not. While acronyms like MIC (line 66) and SOC (line 81) are well known amongst microbiologists, given the wider scope of this paper it might be helpful to give the full definitions of these when first mentioned.
Response 4: We have added the definition of acronyms ‘MIC’ on line 87 and ‘SOC’ on line 76.
Point 5: S. maltophilia on lines 71, 191, 201, 202 and 125 (there may be other instances too) should be in italics.B. xylophilus on page 213 should be in italics.
Response 5: These have been corrected.
Point 6: On line 199 Xing Ye is referred to without a reference. It also doesn’t appear to be given in the References section.
Response 6: Reference for Xing Ye on line 233 has been added to the References section.
Point 7: There are a number of typos in the manuscript: ‘processe’ on line 48, ‘roducing’ on line 235, on lines 205 and 206 the word ‘of’ is missing before ‘usable antibiotics’ and ‘trimethoprim’ respectively.
Response 7: We have checked and corrected typos in the manuscript.
Reviewer 2 Report
Its a nice study planned well and executed. No problem to methodology except the reference in the methodology section are missing. Technical names like E. coli should be italicized throughout the manuscript. In figures footnote,mentioned the standard bars given are SE (Std. error) or SD(Std. deviation). Figure 2: give the SD or SE.
in text, referring results mention the SE/SD with the results value.
Author Response
Thanks a lot to your valuable commentsThe point-to-point responds to the comments are listed as follows:
Point 1: No problem to methodology except the reference in the methodology section are missing.
Response 1: We have added references for methodology in the section ‘2.2 Antimicrobial minimum inhibitory concentration (MIC) test’, ‘2.3 Preparation of competent cells for electroporation’ and ‘2.4 Electroporation’.
Point 2: Technical names like E. coli should be italicized throughout the manuscript.
Response 2: These have been corrected.
Point 3: In figures footnote, mentioned the standard bars given are SE (Std. error) or SD(Std. deviation). Figure 2: give the SD or SE. in text, referring results mention the SE/SD with the results value.
Response 3: The experimental data about Figure 2 is MIC test with gradiently diluted antibiotics ranging from 100ug/ml to1000 ug/ml. Since the replicates gave very consistent results, we therefore consider that SD(Std. deviation) are not necessary to be indicated. So we didn’t modify Figure 2.